# TSC-Net: Prediction of Pedestrian Trajectories by Trajectory-Scene-Cell Classification

**Bo Hu, Tat-Jen Cham**
College of Computing and Data Science
Nanyang Technological University
50 Nanyang Ave, Block N4, Singapore
`hubo0005@e.ntu.edu.sg, astjcham@ntu.edu.sg`

## Abstract

To predict future trajectories of pedestrians, scene is as important as the history trajectory since i) scene reflects the position of possible goals of the pedestrian ii) trajectories are affected by the semantic information of the scene. It requires the model to capture scene information and learn the relation between scenes and trajectories. However, existing methods either apply Convolutional Neural Networks (CNNs) to summarize the scene to a feature vector, which raises the feature misalignment issue, or convert trajectory to heatmaps to align with the scene map, which ignores the interactions among different pedestrians. In this work, we introduce the trajectory-scene-cell feature to represent both trajectories and scenes in one feature space. By decoupling the trajectory in temporal domain and the scene in spatial domain, trajectory feature and scene feature are re-organized in different types of cell feature, which well aligns trajectory and scene, and allows the framework to model both human-human and human-scene interactions. Moreover, the Trajectory-Scene-Cell Network (TSC-Net) with new trajectory prediction manner is proposed, where both goal and intermediate positions of the trajectory are predict by cell classification and offset regression. Comparative experiments show that TSC-Net achieves the SOTA performance on several datasets with most of the metrics. Especially for the goal estimation, TSC-Net is demonstrated better on predicting goals for trajectories with irregular speed.

## 1 Introduction

The problem addressed in this work is that of predicting pedestrian trajectories in the future given historical and scene information Liang et al. (2020a); Deo & Trivedi (2020); Chen et al. (2021a), which has strong technical relevance in many applications, such as self-driving vehicles and human-robot interaction Gulzar et al. (2021); Bennewitz et al. (2002). While there have been promising advances in research in the past few years, this task remains challenging. The behavior of a person is determined by both self-intention and social interaction Hu & Cham (2022). In the context of pedestrians, we consider self-intention to determine the planned destination and intended route, governed by the individual's goal and prior knowledge of the environment, and is a longer term attribute. Social interaction relates to shorter term reactive behavior, such as avoiding collision with other independent pedestrians, or more complex behavioral dynamics such as group movement. Since it is not possible to discern the exact intention of a pedestrian based purely on past visual observations, the prediction of motion becomes less certain the further into the future the prediction is attempted. Therefore, the appropriate approach in this situation is to model the distribution of goals (*i.e.* , intended destinations), together with the distribution of paths to get there. If the environment is known, say in the form of a scene map, this will influence the distribution of goals and paths, *e.g.* , a store entrance will be more probable as a goal than the exterior wall next to it, and paths are more likely to follow walkways. We can consider these effects to be human-scene interactions, as a counterpart to human-human interactions.

Most of the earlier deep learning-based methods for human trajectory prediction had been predominantly focused on human-human interactions Gupta et al. (2018); Alahi et al. (2016); Zhang et al. (2019); Yu et al. (2020). Later works further considered how semantic information from the scene

also significantly affect human decisions, and incorporated scene information into this task. The most intuitive way is to concatenate a scene feature vector obtained from a Convolutional Neural Network (CNN). Both local scene maps Salzmann et al. (2020); Liang et al. (2019) and global maps Sadeghian et al. (2019); Fang et al. (2020) have been investigated as the prediction input in different methods. In these methods, local scene maps are found not to be suitable for longer range inference. For example, the model may not have sufficient input to correctly predict that a pedestrian is heading for a bus stop, if the bus stop is outside the range of the input scene maps, and there is no additional information available. In contrast, the global scene feature contains all necessary information; however, trajectory features are local representations focusing on specific positions. The concatenation of trajectory feature and global scene feature will result in difficulty learning the relations between the trajectory and the scene, as the scale of these two feature types are misaligned. In Mangalam et al. (2021), an alternate strategy is provided to fuse scene maps and trajectories, where the trajectory is converted to pedestrian position-based heatmaps at different time steps, and the heatmaps are concatenated to the scene maps. In this way, the coordinate information is well aligned with the scene semantic information in the spatial domain. However, each pedestrian is represented by a 3-dimensional tensor in Mangalam et al. (2021), which makes the computational cost of computing human-human interactions very high.

In this work, we propose a novel trajectory and scene feature fusion technique. As trajectory information is more dominant in the temporal domain while scene semantic information is more so in the spatial domain, to align these two feature types, trajectory and scene features are decoupled in temporal and spatial domains respectively and re-organized into spatial-temporal cells, which we call Trajectory-Scene-Cell (TSC) features. Every cell feature consists of a coordinate feature and the corresponding local scene feature. The raw trajectory and scene inputs are converted to three types of TSC features: trajectory cells, goal scene cells, and step scene cells. Based on the TSC features, we introduce a new two-stage trajectory prediction framework, the Trajectory-Scene-Cell Network (TSC-Net). Different from directly regressing the coordinates of goal and future trajectory, our TSC-Net completes the trajectory prediction by step-by-step cell classification. The goal prediction network assigns a confidence score to each goal scene cell, of whether the goal is in the cell or not. To obtain a more accurate position, a positional offset from the cell center is also estimated for each cell. Similarly, the confidence and offsets are computed for every step scene cell in the trajectory completion network. The TSC features unify the trajectory feature and scene feature to the same feature space, which keep good alignment between trajectory and scene as trajectory cells and scene cells are represented in the same format at the same scale. Moreover, TSC features allow unification of human-human interaction and human-scene interaction modeling, where all the interactions are modeled by cell-level attentions in our framework.

The main contributions of this paper are as follows:

- Based on a well-reasoned analysis of requirements for integrating trajectory and scene features, we propose a novel joint representation, the *trajectory-scene-cell feature*.
- We introduce an innovative trajectory prediction pipeline, in which *trajectory prediction is achieved via cell classification and sampling*, as enabled by our trajectory-scene-cell feature.
- We demonstrate our approach outperforms most of existing methods in two datasets.

## 2 RELATED WORK

### 2.1 SEQUENCE MODELING

Sequence modeling is a fundamental component of the trajectory prediction task. To encode trajectory histories, Recurrent Neural Networks (RNN) based models Zhang et al. (2019); Kosaraju et al. (2019); Bartoli et al. (2018) and transformers with attention Bae et al. (2024); Hu & Cham (2022); Pang et al. (2021) are most commonly used. Some other methods directly concatenate history coordinates and employ a CNN or Multi-Layer Perceptron (MLP) as the encoder Mangalam et al. (2021; 2020), which is efficient but hard to generalize to different trajectory lengths. Beside the temporal encoder, the decoder and prediction parts also have different designs. Step-by-step prediction can be achieved by an auto-regressive structure, usually realized by RNN Alahi et al. (2016); Zhang et al. (2019); Kosaraju et al. (2019) and attention Yu et al. (2020); Hu & Cham (2022). In

Gu et al. (2022), all future steps are predicted simultaneously, while a diffusion model Ho et al. (2020) can be employed to refine the future trajectory iteratively. Recently, Hierarchical prediction achieves state-of-the-art performance on many datasets. In these methods, prediction of the future trajectory is in two stages Mangalam et al. (2020); Yue et al. (2022), where the final step (goal) is first predicted and used as the guidance when predicting the intermediate steps. In Mangalam et al. (2021), the future trajectory is further divided into goals, waypoints, and paths, to allow for longer term prediction. Liang et al. (2019) provided a gridding mechanism for rough goal prediction as an auxiliary loss over conventional RNN framework. Different from Liang et al. (2019), our TSC-Net predicts precise goal and trajectory purely on cell classification mechanism.

## 2.2 HUMAN-HUMAN INTERACTION

Future trajectory is determined by self-intention and social interaction, with the latter requiring human-human relation modeling. The distance between two pedestrians is used as relation in Gupta et al. (2018), pooling all relations from a neighbor set to a feature vector. A social physics function can also be employed Yue et al. (2022), where the state of every pair of pedestrians is sent to a module to compute repulsion. A higher-order graph is introduced in Kim et al. (2024) to capture collision-aware relations of pedestrians. The success of the transformer Vaswani et al. (2017) led to attention becoming the most popular way to model relations among pedestrians. In Zhang et al. (2019), attention-based human-human interaction modeling is integrated into RNN-based trajectory prediction. In contrast, some other works employed attention to model human-human interaction (spatial relation) and history encoding (temporal relation) simultaneously. In Hu & Cham (2022), a spatial transformer and a temporal transformer are applied sequentially to trajectory history, while other structures such parallel spatial and temporal transformers are explored in Yu et al. (2020).

## 2.3 HUMAN-SCENE INTERACTION

None of the models can accurately predict the intention of a specific person. However, models can be applied to learn the conditional distribution of goals for multiple people; here, scene semantic information is critical, especially for longer-term prediction. A backbone network for global scene feature embedding is employed in Sadeghian et al. (2019), while Dendorfer et al. (2021) took the cropped global scene as input which covers all steps of a trajectory. Global scene features and trajectory features cannot be aligned well by direct concatenation, as will be discussed in Section 3.2. In Liang et al. (2019), local scene features surrounding every frame of the trajectory are extracted. Local scene feature is useful for short-term obstacle avoidance, yet cannot provide clues for long-term goal prediction. Heatmap-based methods have been proposed Mangalam et al. (2021); Cao et al. (2020), where trajectory coordinates are converted to multi-channel heatmaps before concatenated to global scene features. Similarly, Lee et al. (2022) applies heatmaps with a coarse-to-fine structure. Liang et al. (2020b) applied discrete heatmaps that recorded the trajectory with a grid. Different from commonly used Euclidean space, Wong et al. (2024) models the interactions in a angle-based space. Although heatmap-based methods can learn good correspondence between the trajectory and scene, they cannot model human-human interactions because of the high dimensionality.

## 2.4 MULTI-MODALITY PREDICTION

As the indeterminacy of a person's intention is undeniable, many stochastic prediction approaches have been proposed to capture the multi-modality of future trajectories especially the goals. Huang et al. (2019); Yu et al. (2020) directly added noise to the hidden states of the framework to obtain uncertainty, where the best output among multiple samples is selected to compute loss. Instead of direct noise injection, several generative models are applied on this task. Sun et al. (2020); Zhao et al. (2019) employ generative adversarial networks (GANs) Goodfellow et al. (2014) to model the indeterminacy with a noise variable. Some other methods Chen et al. (2021b); Ivanovic & Pavone (2019); Tang & Salakhutdinov (2019); Yue et al. (2022); Bhattacharyya et al. (2019) combined Variational Auto-Encoder (VAE) or conditional VAE (CVAE) Sohn et al. (2015) in their trajectory prediction pipeline. Gu et al. (2022) applied a diffusion model Ho et al. (2020) to generate future positions from noise variables, which achieved good performance on a large dataset but has high computational cost. Lee et al. (2017) sampled future trajectories from latent space, which is optimized by a inverse reinforcement learning module.

## 3 METHOD

### 3.1 PROBLEM DEFINITION

In this task, there are two types of information: trajectories represented by coordinates, and the scene represented by RGB images or semantic segmentation maps. The video clip has $T$ frames and $N$ pedestrians ($N$ varies for different video clip samples). The trajectory of the $i^{\text{th}}$ pedestrian is represented by $T$ 2D coordinates and denoted by $p_{i,t} = (x_{i,t}, y_{i,t})$, where $i = 1, \ldots, N$ and $t = 1, \ldots, T$. The time invariant scene in the video clip is denoted by $\mathcal{I}$, where $\mathcal{I} \in \mathbb{R}^{C \times H \times W}$. Note that the trajectory and scene need to be described in the same coordinate system, *e.g.*, the scenes are RGB images represented in pixel space, and the trajectory coordinates are also in pixels. Given the scene $\mathcal{I}$ and observed trajectories of every pedestrian $\mathbf{p}^{(in)} = \{p_{i,t}\}_{i=1:N, t=1:\tau}$, where $1 \leq \tau < T$, this task is to predict the future trajectories of all pedestrians $\mathbf{p}^{(out)} = \{p_{i,t}\}_{i=1:N, t=\tau+1:T}$.

### 3.2 MULTI-MODEL FEATURE ALIGNMENT

Here, we explore and analyze the alignment of coordinate-based trajectory features and image-based scene features. To combine two different types of features, an intuitive way is concatenating both types of feature embedding Sadeghian et al. (2019); Liang et al. (2019). Usually, separate feature extraction networks are employed for the two feature types, so

$$\mathbf{z}_i = \left[ \mathbf{z}_i^{(co)}, \mathbf{z}_i^{(sc)} \right] = \left[ f\left( \mathbf{p}_i^{(in)} \right), g\left( \mathcal{I} \right) \right], \tag{1}$$

where $f(\cdot)$ and $g(\cdot)$ are feature extraction networks for trajectory coordinates and the scene image respectively. The dimensions in $\mathbf{z}_i^{(co)}$ provide information on the pedestrians' locations. Different dimensions in $\mathbf{z}_i^{(sc)}$ encode semantic information in different areas of the scene, such as the presence of obstacles or sidewalks. To build the relation between pedestrians and scene, the network needs to find specific dimensions of $\mathbf{z}_i^{(sc)}$ based on the content of $\mathbf{z}_i^{(co)}$. However, for different steps or different pedestrians, the relations between dimensions inside $\mathbf{z}_i$ are not fixed. This requires the network to have dynamic weights for different dimensions inside a feature vector, which is difficult for almost all operations, such as convolution, MLP, and attention. Thus, directly concatenating the two types of feature cannot align trajectory and scene information well.

Instead of concatenation, it was proposed in Mangalam et al. (2021) to convert coordinates of a trajectory to heatmaps, where

$$\mathbf{z}_i = \left[ \mathbf{z}_i^{(co)}, \mathbf{z}_i^{(sc)} \right] = \left[ h\left( p_{i,1} \right), \cdots, h\left( p_{i,\tau} \right), \mathcal{I} \right]. \tag{2}$$

In Eq.equation 2, $h(\cdot)$ transforms 2D coordinates to heatmaps with size of $H \times W$, where a Gaussian peak is located at the input coordinates. Therefore, after concatenating the scene maps, the size is $\mathbf{z}_i \in \mathbb{R}^{(C+\tau) \times H \times W}$. At each spatial location, for the feature $\mathbf{z}_{i,j}$, where $j = 1, \ldots, H \times W$, the content to be conveyed is more straightforward than in feature embedding concatenation. More specifically, $z_{i,j}^{(co)}$ should capture the presence or absence of a pedestrian at position $j$, while $z_{i,j}^{(sc)}$ encodes semantic information at the same location. With this structure, the relation between the two feature types is static in different dimensions and can be easily manipulated by simple operations such as convolution, as dynamic dimension selection is unnecessary. Therefore, the key to a good feature alignment is that $\mathbf{z}_i^{(co)}$ and $\mathbf{z}_i^{(sc)}$ should have the same scale and fixed relation. While heatmaps achieve good feature alignment, a pedestrian in Mangalam et al. (2021) is represented by a tensor, which makes social interactions among different pedestrians much harder to compute. Hence, we propose the TSC feature representation, which is able to model social interactions, as well as keep good feature alignment.

### 3.3 TRAJECTORY-SCENE-CELL FEATURES

To achieve good feature alignment, the receptive fields of the two feature types should be small enough and matched to each other. However, a trajectory mainly spans the temporal domain while the scene map spans the spatial domain, To align these two feature types, we propose the trajectory-scene-cell (TSC) feature representation, where both the trajectory and scene are mapped to a unified

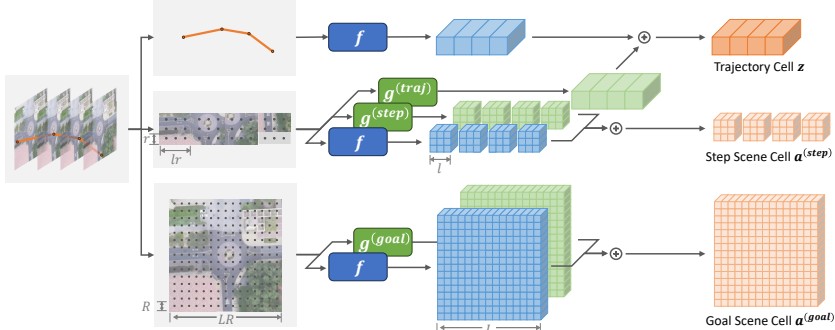

Figure 1: Illustration of the cell feature embedding. $\oplus$ denotes the concatenation. The raw trajectory and scene inputs are converted to three types of TSC feature in the feature embedding stage.

cell feature space. More specifically, the trajectory is decoupled in temporal domain and converted to a sequence of **trajectory cell** feature $\mathbf{z}$ by combining corresponding scene information to each trajectory step. The scene is decoupled to spatial grid and converted to **goal scene cell** feature $\mathbf{a}^{(goal)}$ and **step scene cell** feature $\mathbf{a}^{(step)}$ by integrating center coordinate information to each position in the grid, which is illustrated in Figure 1.

### 3.3.1 TRAJECTORY CELL FEATURE.

To compute $\mathbf{z}$, the trajectory is decoupled into separate temporal cells $z_{i,t}^{(co)}$, for which each cell only represents the coordinates of one time step, *i.e.* ,

$$z_{i,t}^{(co)} = f\left([p_{i,\tau}, \bar{p}_{i,t}, \dot{p}_{i,t}]\right), \tag{3}$$

where the coordinate feature embedding network $f(\cdot)$ is an MLP. It takes in three components of: 2D reference coordinates $p_{i,\tau}$, 3D shifted coordinates $\bar{p}_{i,t}$, and 2D velocity $\dot{p}_{i,t}$. The reference position $p_{i,\tau}$ is the most recent *observed* frame at time $\tau$, while the shifted coordinates $\bar{p}_{i,t}$ is the relative position compared to the reference position, together with the time difference in a third dimension:

$$\bar{p}_{i,t} = [p_{i,t} - p_{i,\tau}, t - \tau]. \tag{4}$$

The velocity $\dot{p}_{i,t}$ is computed as the difference in coordinates between two neighboring frames, *i.e.* ,

$$\dot{p}_{i,t} = \begin{cases} p_{i,t} - p_{i,t-1}, & t > 1 \\ p_{i,t+1} - p_{i,t}, & t = 1. \end{cases} \tag{5}$$

For each $z_{i,t}^{(co)}$, a scene feature $z_{i,t}^{(sc)}$ is assigned, which comes from an area centered on $p_{i,t}$ with size $u \times u$. It is computed as

$$z_{i,t}^{(sc)} = g^{(traj)}\left(\mathcal{I}_{y_{i,t}-\frac{u}{2}:y_{i,t}+\frac{u}{2}, x_{i,t}-\frac{u}{2}:x_{i,t}+\frac{u}{2}}\right), \tag{6}$$

where $g^{(traj)}(\cdot)$ is a CNN-based scene feature extraction network. Therefore, the pedestrian cell feature $z_{i,t}$ is computed by concatenation of two feature types $z_{i,t} = \left[z_{i,t}^{(co)}, z_{i,t}^{(sc)}\right] \in \mathbb{R}^D$, where $D$ is the embedded feature dimension. Then the feature of observation becomes $\mathbf{z}^{(in)} \in \mathbb{R}^{D \times N \times \tau}$.

### 3.3.2 GOAL SCENE CELL FEATURE.

To map the scene into the same feature space as a trajectory cell feature $\mathbf{z}$, the scene is decoupled into two types of scene cell features, $\mathbf{a}^{(goal)}$ and $\mathbf{a}^{(step)}$. To obtain $\mathbf{a}^{(goal)}$, a square area centered on the last observation position $p_{i,\tau}$ with the size of $LR \times LR$ is cropped (and padded if necessary), followed by a CNN $g^{(goal)}(\cdot)$ maps the area to $L \times L$ cells, where $L$ is the number of goal cells in one axis of the grid and $R$ is the size of one cell. For each cell, the center coordinate feature is assigned. Hence, the goal scene cell feature is computed by concatenating the coordinate features and scene feature:

$$a_{i,j,k}^{(goal)} = \left[f\left([p_{i,\tau}, q_{i,j,k}, T - \tau, q_{i,j,k}]\right), g^{(goal)}\left(\mathcal{I}_i^{(goal)}\right)_{j,k}\right], \tag{7}$$

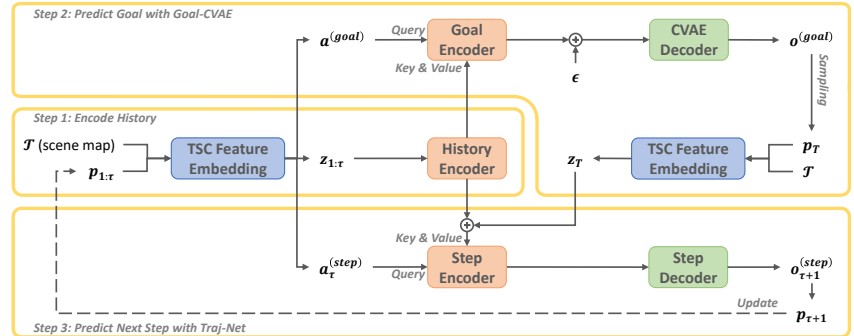

Figure 2: Framework of TSC-Net. 1) Compute TSC features (illustrated in Figure 1) and encode features in historical frames, where the output is used in the next two steps. 2) Predict goals form Goal-CVAE, the goal prediction network with CVAE structure, where the output is used in trajectory completion as a guidance. 3) Predict next step position with Traj-Net, the trajectory completion network. Step 1 and 3 are applied cyclically until all steps are predicted. $\oplus$: concatenation.

where

$$q_{i,j,k} = \left[ \left(k - \frac{L-1}{2}\right) R, \left(j - \frac{L-1}{2}\right) R\right],$$

$$\mathcal{I}_i^{(goal)} = \mathcal{I}_{y_{i,\tau} - \frac{LR}{2} : y_{i,\tau} + \frac{LR}{2}, x_{i,\tau} - \frac{LR}{2} : x_{i,\tau} + \frac{LR}{2}}.$$

(8)

In this way, each pedestrian has a unique but global scene representation $\mathbf{a}^{(goal)} \in \mathbb{R}^{D \times N \times L^2}$.

### 3.3.3 STEP SCENE CELL FEATURE.

Similarly, step scene cell features $\mathbf{a}^{(step)}$ are formed by:

$$a_{i,t,j,k}^{(step)} = \left[ f\left([p_{i,\tau}, q_{i,t,j,k} + p_{i,t} - p_{i,\tau}, t - \tau, q_{i,t,j,k}]\right), g^{(step)}\left(\mathcal{I}_{i,t}^{(step)}\right)_{j,k}\right],$$

(9)

where

$$q_{i,t,j,k} = \left[ \left(k - \frac{l-1}{2}\right) r, \left(j - \frac{l-1}{2}\right) r\right],$$

$$\mathcal{I}_i^{(step)} = \mathcal{I}_{y_{i,t} - \frac{lr}{2} : y_{i,t} + \frac{lr}{2}, x_{i,t} - \frac{lr}{2} : x_{i,t} + \frac{lr}{2}}.$$

(10)

$g^{(step)}(\cdot)$ maps the $lr \times lr$ area to $l \times l$ cells, where $l$ is the number of step cells in one axis of the grid and $r$ is the size of one cell. Step scene cells $\mathbf{a}^{(step)} \in \mathbb{R}^{D \times N \times L^2 \times \tau}$ are built for all time steps, where the cropped scene has a smaller size than goal scene and centered on the position of each time step. To simplify the implementation, we set $u = lr$.

TSC features can be treated as a local scene feature concatenated with a learnable spatial-temporal positional encoding Xu et al. (2021); Dosovitskiy et al. (2020). The TSC feature embedding module provides positional encoding for every local scene and "scene encoding" for every position of the trajectory. Both trajectories and scenes are converted into this format. Therefore, the information inside trajectory cell features and scene cell features are matched with each other and the relation among feature dimensions become static.

### 3.4 PREDICTING TRAJECTORY BY CELL CLASSIFICATION

TSC features enable both human-human and human-scene interactions to be modeled by cell-level attention operations. With help of TSC features, the trajectory prediction task becomes that of learning a mapping from $\left(\mathbf{z}^{(in)}, \mathbf{a}^{(goal)}, \mathbf{a}^{(step)}\right)$ to $\mathbf{p}^{(out)}$. Our framework follows previous works Mangalam et al. (2021) in splitting the prediction into *two stages* of **goal estimation** and **trajectory completion** (Figure 2). Instead of auto-regressive approaches, in our method, the goal and trajectory predictions are completed by estimating the likelihood of each cell containing the goal or the next step. In other words, *the trajectory prediction task is reformulated as a classification problem.*

### 3.4.1 GOAL PREDICTION NETWORK.

In the goal prediction network, the history trajectory feature $z_{1:\tau}$ is first sent to the history encoder, which consists of $K$ self-attention layers over spatial and temporal domains. Both human-human interactions and temporal relations within each trajectory are captured by the history encoder. The goal encoder has $K$ self-attention layers for $\mathbf{a}^{(goal)}$, and also $K$ cross-attention layers for attention between $\mathbf{a}^{(goal)}$ and the output of the history encoder $\tilde{z}_{1:\tau}$, where $\mathbf{a}^{(goal)}$ is the query since the target is to predict the presence of the goal in each goal scene cell. Note that self-attentions and cross-attentions in the goal encoder are only computed with the same pedestrian, as human-human interaction information is already captured in the history encoder. A CVAE structure Kingma & Welling (2013) is applied for multi-modality goal prediction, where the ground truth embedding, CVAE encoder, and CVAE decoder are all cell-level MLPs. The condition of CVAE is the output of goal encoder $\tilde{\mathbf{a}}^{(goal)}$. After combining the condition and latent variable $\epsilon$, CVAE estimates the goal output $\mathbf{o}^{(goal)} \in \mathbb{R}^{3 \times N \times L^2}$. For each cell, $o_{i,j,k}^{(goal)} \in \mathbb{R}^3$ includes the 1-dimension confidence of whether the goal is located in the cell and the 2D offset between the goal and cell center.

### 3.4.2 TRAJECTORY COMPLETION NETWORK.

After sampling the goals $p_T$, TSC features $z_T$ are computed, which represent the guidance when predicting the trajectory. The trajectory completion network shares the same encoder-decoder structure except CVAE. When predicting the trajectory in frame $t+1$, the step encoder first computes the self-attention for $a_t^{(step)}$ and then computes the cross-attention over $a_t^{(step)}$ and $\tilde{z}_{1:t}$, with the query for cross-attention being $a_t^{(step)}$. The output of the step encoder $\tilde{a}_t^{(step)}$ is sent to the step decoder, a cell-level MLP, to predict the step output of frame $t + 1$: $\mathbf{o}_{t+1}^{(step)} \in \mathbb{R}^{3 \times N \times l^2}$. Here $o_{t+1,i,j,k}^{(step)}$ also includes one confidence score and 2 offsets. The position of next step $p_{t+1}$ is determined by picking the cell with highest confidence and adding the corresponding offsets to the cell center. Given $p_{t+1}$, $\mathcal{I}_{t+1}^{(step)}$ is cropped, $a_{t+1}^{(step)}$ computed, and $z_{1:t+1}$, $\tilde{z}_{1:t+1}$ updated by the feature embedding module and history encoder.

The loss function of TSC-Net includes the CVAE loss from goal network and a L2 reconstruction loss from trajectory completion network:

$$L = D_{KL}(q(\epsilon|\mathbf{o}^{(goal)}, \tilde{\mathbf{a}}^{(goal)})\|p(\epsilon)) + \lambda\|\mathbf{o}^{(goal)} - \hat{\mathbf{o}}^{(goal)}\|_2 + \alpha\|\mathbf{o}^{(step)} - \hat{\mathbf{o}}^{(step)}\|_2, \quad (11)$$

where $\lambda$ and $\alpha$ are weights for different parts, $\epsilon$ is the latent variables of CVAE, and $D_{KL}(\cdot\|\cdot)$ is the Kullback–Leibler divergence of two distributions.

### 3.5 HYBRID GOAL SAMPLING

In our model, CVAE is employed to generate multiple trajectories. However, different from previous methods, the goal CVAE network in our model naturally have more sampling solutions as the output of CVAE are cells with confidence scores. In the inference stage, for each pedestrian, CVAE first generates $m$ $L \times L$ grids. Secondly, for each grid, $n$ cells with highest confidence score are picked. Hence, $mn$ predicted goals are obtained from this two-step sampling.

In practical, there are multiple $(m, n)$ combinations, where following widely used setting $mn = 20$. We find that larger $m$ and smaller $n$ is likely to result in goals with small variance, which will achieve better performance for short or straight trajectories. Conversely, smaller $n$ and larger $m$ tends to larger variance, which is helpful to longer complex trajectories. Hence a hybrid sampling strategy is proposed to find the most appropriate $(m, n)$ combination for each pedestrian.

During training, a hyper classifier is trained separately, where input is the $p_{i,1:\tau}$ for a pedestrian, and category label is the $(m, n)$ combination index that achieves the best performance. In the inference stage, a pedestrian specific $(m, n)$ combination is obtained from the classifier before run the TSC-Net. The hyper classifier can be treated as a model selection module that determines the hyper-parameters $m$ and $n$ for each pedestrian. The hyper classifier is built by an MLP and easy to train since the relation between trajectory shape and best $(m, n)$ combination is stable and general over training and testing sets.

# 4 EXPERIMENTS

## 4.1 DATASETS AND METRICS

Stanford Drone Dataset (SDD) Robicquet et al. (2016) is a large-scale benchmark which contains more than 11,000 pedestrians with 20 different scenes. The dataset is pre-processed for the trajectory prediction following NSP-SFM Yue et al. (2022). Intersection Drone Dataset (InD) Bock et al. (2020) contains about 10,000 pedestrians in 4 different road intersection scenes. ETH-UCY dataset Lerner et al. (2007); Pellegrini et al. (2009) includes 5 subsets, where the evaluation follows the leave-one-out validation strategy over 5 subsets. The scene semantic maps in this dataset have 2 category labels: traversable area and non-traversable area.

**Prediction Settings** in the experiments include short-term prediction and long-term prediction. Most of previous works focus on the short-term setting with $T=20$ and $\tau=8$, where the source videos are down-sampled to 2.5 fps. The short-term setting is applied in the experiments on SDD and ETH-UCY datasets. Following Mangalam et al. (2021), we apply the long-term setting $T=35$ and $\tau=5$ with 1 fps frame rate, which is applied in the experiment on SDD and inD.

**Metrics** used in the experiments include Average Displacement Error (ADE) and Final Displacement Error (FDE) are measured Alahi et al. (2014), where ADE is the average distance between prediction and ground truth over all frames and FDE is the distance between prediction and ground truth of the last frame. With multi-modality prediction, minimum ADE and FDE over 20 sampled trajectories for each person are presented.

## 4.2 ABLATION STUDY

We investigated several alternate selections for different parts of our model in SDD, where results are shown in Table 1. First, multiple sizes of the goal scene cells were evaluated, which is denoted as different $L\times L$. $L=15$ outperforms both smaller and larger $L$ values. Smaller $L$ means the network has to find correct offsets in a larger area, while larger $L$ means the network has to classify the correct cell among more cells. $L=15$ balances the number of cells and the area of each cell.

Different two-step sampling hyper-parameters were also evaluated. There are 6 different $(m, n)$ combinations that get 20 samples. The selection of $m$ and $n$ determined the range and precision of the goal sampling. Usually short and straight observed trajectories achieved better prediction performance with larger $m$, where more goals can be sampled in the same cell. In contrast, long and irregular observed trajectories achieved better prediction performance with larger $n$, which force the sampled goals to be distributed in a larger area. Since hybrid sampling tries to find the most suitable $(m, n)$ combination for each pedestrian, it achieves the least ADE and FDE compared to all fixed $(m, n)$ combinations. $(10, 2)$ has the second best ADE and FDE, which is a balance between the precision and range over all trajectories.

Table 1: Ablation study results in SDD dataset.

| Input | -Scene | -$\bar{p}_{i,t}$ | -$p_{i,\tau}$ | -$\dot{p}_{i,t}$ | Full | | | Full | | | | | | Full | |
|---|---|---|---|---|---|---|---|---|---|---|---|---|---|---|---|
| $L\times L$ | 15×15 | | | | 1×1 | 5×5 | 25×25 | 15×15 | | | | | | 15×15 | |
| (m,n) | Hybrid | | | | (20,1) | Hybrid | | (20,1) | (10,2) | (5,4) | (4,5) | (2,10) | (1,20) | Hybrid | |
| Query | Scene | | | | Scene | | | Scene | | | | | | Traj | Scene |
| ADE | 8.33 | 8.31 | 7.10 | 15.28 | 8.77 | 7.49 | 9.70 | 8.35 | 7.18 | 7.49 | 7.66 | 8.43 | 9.63 | 19.58 | **6.44** |
| FDE | 13.93 | 13.99 | 11.34 | 14.66 | 14.70 | 11.92 | 17.01 | 14.53 | 11.78 | 12.64 | 13.10 | 14.85 | 17.45 | 27.20 | **9.97** |

Moreover, to demonstrate the effectiveness of prediction by cell classification (denoted as "Scene"), we designed an alternate goal encoder and step encoder with a common auto-regressive structure. For the alternate encoders, the last frame trajectory cell was sent as query and the decoders were trained to directly regress the goal (or next frame) coordinates, which is denoted as "Traj". The results show cell classification significantly outperforms the auto-regressive version. We believe this is because cell classification is easier to learn with the format of TSC features. In cell classification, one scene cell is responsible for a small area. However, when directly predicting the next step position, the network has to regress too many different possible areas based on the last pedestrian

cell, which is more difficult. At last, different input components are compared by excluding one input feature type at one time (*e.g.*, '-Scene' indicates the full input types without scene). It can be observed that the velocity $\dot{p}_{i,t}$ plays the most crucial role during prediction.

## 4.3 COMPARISON WITH STATE-OF-THE-ART METHODS

Here, our method is quantitatively compared with a wide range of existing methods, with results shown in Table 2. For SDD, our TSC-Net outperforms SOTA methods NSP-SFM Yue et al. (2022) by 0.64 on FDE and 0.08 on ADE. Note that heatmap based methods NSP-SFM and Y-Net Mangalam et al. (2021) requires 10,000 samples in testing for every pedestrian before obtain the final 20 samples by clustering, where the samples can naturally cover larger possible positions. In contrast, our hybrid sampling strategy only needs to sample more goals in the training stage to learn the hyper-classifier. Our performance also significantly surpasses scene feature embedding based methods, such as SoPhie Sadeghian et al. (2019) and MG-GAN Dendorfer et al. (2021).

Results from ETH-UCY dataset are also presented. The scene semantic maps in ETH-UCY dataset are only binary labels: traversable and untraversable and usually the whole scene is traversable. This meant that there is little relationship between the scene and trajectory features to be captured by TSC features, thus, in this scenario our TSC features degenerate to normal trajectory feature embeddings and the performance is limited by inadequate scene maps. From the results it can be seen that TSC-Net achieves the third lowest FDE on the average of all subsets except for the two methods using 10,000 samples. The results demonstrate that the cell classification manner has comparable performance even with normal feature embedding only.

Table 2: Comparison between our method and existing methods on ETH-UCY dataset and SDD dataset. "ADE/FDE" are reported. * means the results come from clustering 10,000 samples, where all other methods require 20 samples only. **Bold**:Best, Underline:Second best.

| Methods | ETH | HOTEL | UNIV | ZARA1 | ZARA2 | Average | SDD |
|---|---|---|---|---|---|---|---|
| Y-Net* Mangalam et al. (2021) | 0.28/0.33 | 0.10/0.14 | 0.24/0.41 | 0.17/0.27 | 0.13/0.22 | 0.18/0.27 | 7.85 /11.85 |
| NSP-SFM* Yue et al. (2022) | 0.25/0.24 | 0.09/0.13 | 0.21/0.38 | 0.16/0.27 | 0.12/0.20 | 0.17/0.24 | 6.52 /10.61 |
| Social-GAN Gupta et al. (2018) | 0.81/1.52 | 0.72/1.61 | 0.60/1.26 | 0.34/0.69 | 0.42/0.84 | 0.58/1.18 | 27.23/41.44 |
| SoPhie Sadeghian et al. (2019) | 0.70/1.43 | 0.76/1.67 | 0.54/1.24 | 0.30/0.63 | 0.38/0.78 | 0.54/1.15 | 16.27/29.38 |
| Trajectron++ Salzmann et al. (2020) | 0.39/0.83 | 0.12/0.21 | **0.20**/0.44 | **0.15**/0.33 | **0.11**/0.25 | 0.19/0.41 | 8.98 /19.02 |
| MG-GAN Dendorfer et al. (2021) | 0.47/0.91 | 0.14/0.24 | 0.54/1.07 | 0.36/0.73 | 0.29/0.60 | 0.36/0.71 | 13.60/25.80 |
| PCCSNET Sun et al. (2021) | 0.28/0.54 | **0.11**/0.19 | 0.29/0.60 | 0.21/0.44 | 0.15/0.34 | 0.21/0.42 | 8.62 /16.16 |
| MID Gu et al. (2022) | 0.39/0.66 | 0.13/0.22 | 0.22/0.45 | 0.17/0.30 | 0.13/0.27 | 0.21/0.38 | 7.61 /14.30 |
| FlowChain Maeda & Ukita (2023) | 0.55/0.99 | 0.20/0.35 | 0.29/0.54 | 0.22/0.40 | 0.20/0.34 | 0.29/0.52 | 9.93/17.17 |
| SICNet Dong et al. (2023) | 0.27/0.45 | **0.11**/0.16 | 0.26/0.46 | 0.19/0.33 | 0.13/0.26 | 0.19/0.33 | 8.44/13.65 |
| TUTR Shi et al. (2023) | 0.40/0.61 | **0.11**/0.18 | 0.23/0.42 | 0.18/0.34 | 0.13/0.25 | 0.21/0.36 | 7.76/12.69 |
| V$^2$-Net Wong et al. (2022) | **0.23/0.37** | **0.11**/0.16 | 0.21/0.35 | 0.19/0.30 | 0.14/0.24 | 0.18/0.28 | 7.12/11.39 |
| LMTraj-SUP Bae et al. (2024) | 0.41/0.51 | 0.12/0.16 | 0.22/0.34 | 0.20/0.32 | 0.17/0.27 | 0.22/0.32 | 7.80/10.10 |
| E-V$^2$-Net-SC Wong et al. (2024) | 0.25/0.38 | 0.12/**0.14** | **0.20**/**0.34** | 0.18/**0.29** | 0.13/**0.22** | **0.17/0.27** | 6.54/10.36 |
| TSC-Net (Ours) | 0.32/0.39 | 0.12/0.19 | 0.25/0.46 | 0.17/0.30 | 0.15/0.26 | 0.20/0.32 | **6.44/9.97** |

## 4.4 LONG-TERM PREDICTION

In this section we evaluate our TSC-Net in a long-term prediction setting. Our method achieves best FDE on Both SDD and InD and comparable ADE. With longer prediction range, the size of cell becomes larger, where less scene details are summarized to the representation. Thus, ADE of our method is slightly larger than heatmap based method Y-Net. SocialCircle Wong et al. (2024) achieves good performance on ETH-UCY dataset, however, it becomes difficult to precisely collecting information with angle-based partition.

1,000 sampled goals of Y-Net and our method on SDD are visualized in Figure 3. In our method, goals are obtained by $(m, n) = (50, 20)$. In Y-Net, 999 goals are sampled from the estimated heatmap besides the soft-argmax sample. It can be observed that our method provides a goal distribution covering more possibilities that accounts for sudden changes of velocity, *e.g.* in Figure 3,

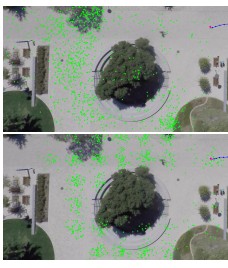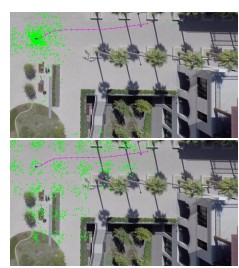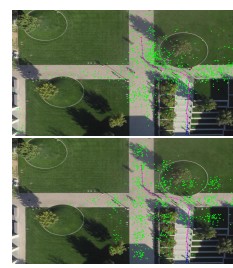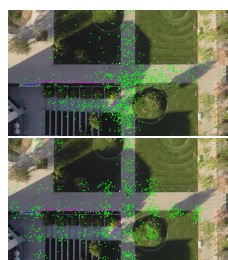

Figure 3: Visualization of trajectory prediction results comparison between Y-Net and our method on SDD dataset with long-term setting. The first row show results from Y-Net, while the second row show results from our method. Blue and magenta curves: history and future of ground truth trajectory. Green dots: predicted goals.

sudden stops (first column) and sudden moves (second column), while Y-Net tends to predict the goals by extrapolating the observed trajectory along the map with a constant speed.

Table 3: Prediction results on SDD dataset and InD dataset with long-term prediction setting, where "ADE/FDE" are reported. †Results are reproduced and reported by Y-Net Mangalam et al. (2021). ‡we re-train the model with long-term setting based on the released code. * The results come from clustering 10,000 samples, where all other methods require 20 samples only.

| Methods | Social-GAN† Gupta et al. (2018) | PECNet† Mangalam et al. (2020) | R-PECNet† | Y-Net* Mangalam et al. (2021) | E-V$^2$-Net-SC ‡ Wong et al. (2024) | TSC-Net (Ours) |
|---|---|---|---|---|---|---|
| SDD | 155.32/307.88 | 72.22/118.13 | 261.27/750.42 | **47.94**/66.71 | 52.16/69.85 | 51.39/**63.97** |
| InD | 38.57/84.61 | 20.25/32.95 | 341.8/1702.64 | **14.99**/21.13 | 16.23/23.56 | 17.15/**19.53** |

## 4.5 LIMITATIONS

Although TSC-Net has various advantages over both heatmap based and scene embedding based methods, and achieves promising performance, the predicted offsets are usually biased to the center of the cell, which makes the sampled goals hard to cover the edge area of cells. It may caused by a non-ignorable amount of pedestrians without any movements in the dataset, as they share the same zero-offset. One possible solution is estimating whether a pedestrian will remain static or move, and sampling the goal separately in two situations. We leave it to future work as integrating such a module is non-trivial.

## 5 CONCLUSIONS

In this work, a novel TSC feature representation was proposed to represent the trajectory and scene in a unified feature space. TSC feature avoids the dynamic weight learning in scene embedding based methods and the high dimensionality issue in heatmap based methods. Both human-human and human-scene interactions can be modeled by cell-level attentions over TSC feature. With TSC features, an innovative framework TSC-Net was introduced, where future positions are predicted by cell classification and offset regression. TSC-Net was evaluated in several experiments and outperformed all compare methods on most of the metrics. Moreover, qualitative results demonstrated TSC-Net generates more reasonable distributions according to the training samples. Source code of the TSC-Net is released at github.com/hubovc/TSC-Net.

## ACKNOWLEDGMENTS

This study is supported under the RIE2020 Industry Alignment Fund – Industry Collaboration Projects (IAF-ICP) Funding Initiative, as well as cash and in-kind contribution from Singapore Telecommunications Limited (Singtel), through Singtel Cognitive and Artificial Intelligence Lab for Enterprises (SCALE@NTU).

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
