# OpenReview forum: "TSC-Net: Prediction of Pedestrian Trajectories by Trajectory-Scene-Cell Classification"
_ICLR.cc/2025/Conference — ICLR 2025 Poster_

### Official Review · Reviewer_fgC6 · 2024-10-27

**Soundness:** 3
**Presentation:** 2
**Contribution:** 2
**Rating:** 5
**Confidence:** 4

**Summary:**

The authors emphasize the importance of utilizing scene information for trajectory prediction and propose a method to do so.
According to the authors, existing approaches have limitations: some methods use CNNs to convert scenes into feature vectors, leading to misalignment issues, while others employ heatmaps to represent trajectories as image features, neglecting pedestrian interactions.
This paper introduces "trajectory-scene-cell feature" to represent both the trajectory and the scene with the same representation.
By integrating these two types of information into a single representation, the method enables human-human and human-scene interactions.
To decode future trajectory from this representation, a Trajectory-Scene-Cell Network is proposed.
It achieves state-of-the-art prediction performance on some datasets and some metrics.

**Strengths:**

* Previous pedestrian trajectory prediction papers tend not to emphasize the importance of effectively utilizing scene information, but this paper points out this gap.
* The paper is well-structured and easy to follow.
* The proposed method is intuitively understandable, technically well-implemented, and achieves comparable prediction performance.

**Weaknesses:**

* The referenced trajectory prediction papers focus heavily on older works. It would be beneficial to include more recent papers.
* If the authors claim that the ETH-UCY dataset only contains binary-level semantics, why not experiment with vehicle trajectory datasets instead? Vehicle datasets offer richer semantic scene information, such as lanes, road edges, and traffic lights, so it seems more convincing to conduct experiments on vehicle datasets for claiming effective use of scene information. Moreover, numerous methods already proposed how to effectively utilize these scene information:
    - HiVT (CVPR, 2022): It proposes an integrated hierarchical transformer structure using both scene and trajectory vectors.
    - Leveraging Future Relationship Reasoning for Vehicle Trajectory Prediction (ICLR, 2023): It models agent interactions using vectorized scene information.
    - Apart from these vectorized methods, many models also use image features. Demonstrating state-of-the-art performance in vehicle scenarios by leveraging scene information would be more convincing.
* The CVAE-based trajectory decoder for goal prediction does not seem new. The sequential prediction of goals and trajectories has been widely explored. For instance, Multimodal Trajectory Prediction Conditioned on Lane-Graph Traversals (CoRL, 2022) presents a more sophisticated approach with predictions conditioned on traversals.
* The ablation study sounds insufficient. It is hard to verify the effectiveness of the proposed TSC. There is only one ablation experiments, but it is hard to see where the performance gain came from (from the proposed methods or finetuning hyperprameters). For example, it is necessary to observe how prediction performance changes when certain features ($z_{i,t}^{(sc)}$ or $a_{i,t,j,k}^{(step)}$) are excluded.
* There are other methods with better performance on the ETH-UCY dataset, making it difficult to claim state-of-the-art performance based on average metrics.
    - View Vertically (ECCV, 2022): 0.18/0.28

**Questions:**

* The ETH-UCY dataset also includes RGB images of the scene. Instead of using binary label semantics, why not use the RGB images? It’s unclear why RGB information was used for the SDD dataset but not for ETH-UCY.
* In equations 7 and 9, is the network 𝑓 the same as the 𝑓 in equation 3? If they are different, the notation should be distinguished. If they are the same, how can they be identical when the inputs in equations 3 and 7 are different?

---

> ### Author Response · Authors · 2024-11-23
> **Rebuttal by Authors**
>
> > Weakness 1 and 5: The referenced trajectory prediction papers focus heavily on older works. It would be beneficial to include more recent papers. There are other methods with better performance on the ETH-UCY dataset, making it difficult to claim state-of-the-art performance based on average metrics.
>
> Response: In the experiment, we compared several papers, including FlowChain, TUTR, SICNet, LMTraj-SUP, E-V$^2$-Net_SC, which are published in 2023, 2024.
>
> We already included 'View Vertically' in our manuscript (V$^2$-Net Wong et al. (2022) in Table 2). Compared with 'View Vertically', we significantly outperform it on SDD dataset. And we didn't claim SOTA performance on ETH-UCY dataset in our manuscript.
>
> > Weakness 2: If the authors claim that the ETH-UCY dataset only contains binary-level semantics, why not experiment with vehicle trajectory datasets instead?
>
> Response: We believe vehicle trajectory prediction is not the same task as pedestrian trajectory prediction, as vehicle prediction involves more prior knowledge, such as traffic rules. Moreover, all the compared baseline methods in our manuscript focus only on the pedestrian scenario, where pedestrian trajectory prediction can sometimes be even more challenging. For example, pedestrian trajectories may include sudden changes in direction and speed, while vehicle trajectories are generally smoother.
>
> > Weakness 3: The CVAE-based trajectory decoder for goal prediction does not seem new. The sequential prediction of goals and trajectories has been widely explored.
>
> Response: We did not claim that CVAE and two-step sequential prediction are the novelties of this work. Our main novelty and contribution lie in proposing a novel joint representation, the TSC feature, which maps trajectory and scene into a unified feature space.
>
> > Weakness 4: The ablation study sounds insufficient. It is hard to verify the effectiveness of the proposed TSC. For example, it is necessary to observe how prediction performance changes when certain features $z_{i,t}^{(sc)}$ or $a_{i,t,j,k}^{(step)}$ are excluded.
>
> Response: We conducted an extensive ablation study in Section 4.2, such as excluding different feature components, changing the goal grid size, altering the sampling strategy, and modifying the prediction manner. Excluding $z_{i,t}^{(sc)}$ means revoming the scene, which is part of the ablation study on excluding different feature components. Excluding $a_{i,t,j,k}^{(step)}$ is not possible, as it serves as the query in the attention operation, which is one of our key designs in the prediction pipeline. However, the ablation on changing the prediction manner provides a comparison to an alternative approach where $a_{i,t,j,k}^{(step)}$ is used as the key and value input.
>
> > Questions 1: The ETH-UCY dataset also includes RGB images of the scene. Instead of using binary label semantics, why not use the RGB images? It’s unclear why RGB information was used for the SDD dataset but not for ETH-UCY.
>
> Response: Following previous methods such as Y-Net, we use semantic scene maps for all the datasets instead of RGB maps. The only difference is that the semantic map in the SDD dataset has five categories, while ETH-UCY has only two categories (traversable and non-traversable, and traversable usually dominates the scene).
>
> > Questions 2: In equations 7 and 9, is the network $f$ the same as the $f$ in equation 3? If they are different, the notation should be distinguished. If they are the same, how can they be identical when the inputs in equations 3 and 7 are different?
>
> Response: Yes, all the coordinate feature encoding $f$ is the same. Equation 3 computes the encoding of trajectory coordinates, while Equation 7 compute the encoding of different cell centers coordiante in the goal grid (Equation 9 encodes the cell centers coordiante in each step grid). The outputs of Equation 3 and Equation 7 are different and play different roles in the prediction pipeline.

---

> > ### Comment · Reviewer_fgC6 · 2024-11-26
> >
> > Thank you for your responses to my questions and comments.
> > I understand your clarification on Weakness 5 and Weakness 3.
> >
> > However, I remain unconvinced that the paper provides sufficient experimental evidence to demonstrate the effectiveness of the proposed method.
> > A significant part of your contribution focuses on leveraging scene information for trajectory prediction.
> > This is why I believe conducting experiments on vehicle trajectory datasets is essential.
> > Richer scene semantics in vehicle datasets, such as lanes, traffic signals, and crosswalks, are inherently tied to traffic rules and provide a better platform to evaluate how well your method can utilize scene information.
> > If your method genuinely excels at incorporating and leveraging semantic scene details, this would be better substantiated through experiments on vehicle trajectory prediction.
> >
> > Regarding the ablation study, I respectfully disagree with your justification for not excluding $a_{i,t,j,k}^{(step)}$.
> > While it is a key component of your method, conducting ablation experiments that exclude it is precisely the point of demonstrating its necessity and effectiveness.
> > By removing $a_{i,t,j,k}^{(step)}$ and observing the impact on performance, you can provide stronger evidence that it plays a critical role in the overall success of your approach.
> > This is a common and essential practice to rigorously validate the effectiveness of a proposed feature or method.

---

> ### Author Response · Authors · 2024-11-27
> **Response to Reviewer fgC6**
>
> Thank you for taking the time to respond to the authors' rebuttal. In this response we will expalin why our method cannot be applied on Argoverse dataset and provide more ablation study experiment results.
>
> > I believe conducting experiments on vehicle trajectory datasets is essential. Richer scene semantics in vehicle datasets, such as lanes, traffic signals, and crosswalks, are inherently tied to traffic rules and provide a better platform to evaluate how well your method can utilize scene information.
>
> We agree that vehicle trajectory datasets like Argoverse provide richer scene semantics. However, scene maps in pedestrian trajectory prediction only indicate 2D images where different channels represent semantic categories. Our method is designed to only accept trajectory data and 2D scene maps as input. Differently, scenes in vehicle trajectory prediction are usually structured information, such as lane and traffic signal data, which cannot be processed by our model. For instance, the scenes in the Argoverse dataset are represented by lane centerlines, traffic direction, and intersection annotations rather than 2D images. As a result, our method cannot be directly applied to this dataset.
>
> Nonetheless, we appreciate the reviewer highlighting the vehicle trajectory prediction task. Given the similar logic between the two tasks, exploring how to adapt our method for vehicle datasets would be an interesting research direction in the future.
>
> > By removing $a_{i,t,j,k}^{(step)}$ and observing the impact on performance, you can provide stronger evidence that it plays a critical role in the overall success of your approach.
>
> Regarding the ablation study on $a_{i,t,j,k}^{(step)}$, we would like to point out that our framework is based on many attention operations, for example, by sending cells in a grid as query and historical trajectory as key and value, the cells summarize its own information with trajectory information so that it can predict whether next step is in the cell or not. Excluding $a_{i,t,j,k}^{(step)}$ means removing the query of an attention function, where the attention cannot get any output.
>
> To evaluate the performance without $a_{i,t,j,k}^{(step)}$, attention functions in the trajectory completion step are removed, where the last step feature is directly sent to the prediction layer to get current step coordinates. Moreover, as $a_{i,j,k}^{(goal)}$ plays a similar role in the goal CVAE, we also modify the Goal CVAE framework in a similar way as excluding $a_{i,t,j,k}^{(step)}$ and evaluate the performance without both $a_{i,j,k}^{(goal)}$ and $a_{i,t,j,k}^{(step)}$. Following our exisiting ablation study, SDD dataset with short-term setting is evaluated. The results are shown in the table below.
>
> | Methods | ADE/FDE |
> |----------|----------|
> | w/o $a_{i,t,j,k}^{(step)}$  | 10.22 / 14.85   |
> | w/o $a_{i,j,k}^{(goal)}$ and $a_{i,t,j,k}^{(step)}$   | 20.82 / 29.54   |
> | Full Model   | 6.44 / 9.97   |
>
> From the table it can be seen that excluding $a_{i,t,j,k}^{(step)}$ results in a larger ADE. Although in this experiment goal CVAE network has not been changed, FDE is still larger than the full model. This is because the goal CVAE and trajectory completion networks are trained in end-to-end manner. If the training loss in trajectory completion network is much larger, the model will put most effort to optimize trajectory completion network and affect the goal CVAE optimization. After excluding both $a_{i,j,k}^{(goal)}$ and $a_{i,t,j,k}^{(step)}$, ADE and FDE is significantly larger than our full model.

---

### Official Review · Reviewer_RVzK · 2024-10-30

**Soundness:** 3
**Presentation:** 2
**Contribution:** 3
**Rating:** 5
**Confidence:** 4

**Summary:**

In this study, the trajectory-scene-cell feature is introduced to integrate both trajectories and scenes into a unified feature space. By decoupling the trajectory in the temporal domain and the scene in the spatial domain, the trajectory and scene features are reorganized into various cell types that effectively align both aspects. This approach facilitates the modeling of interactions between humans as well as interactions between humans and their environments. Additionally, The TSC-Net is introduced, utilizing an innovative trajectory prediction approach that predicts both the goal and intermediate trajectory positions through cell classification and offset regression. Experiments conducted on two datasets demonstrate that TSC-Net improves the accuracy of pedestrian trajectory prediction.

**Strengths:**

1.This paper proposes a novel joint representation called the trajectory-scene-cell feature and introduces an innovative trajectory prediction pipeline.

2.The structure of the paper is comprehensive, and the writing is clear. Compared to the baselines, it demonstrates commendable results on several public datasets.

**Weaknesses:**

1.The layout design of Figure 2 is unappealing and does not effectively highlight the core innovations of this research. It is recommended to make the various steps of the framework more explicit in the figure, such as the model's inputs and outputs.

2.The paper is quite challenging to read due to the confusing formula symbols, such as the Formula 7 and Formula 8, what does $q$ represent for and what is the meaning of $i$, $j$ and $k$?

**Questions:**

1. In Formula 11, what is the prior distribution of $\varepsilon$, and does $p(\varepsilon)$ represent the standardized normal distribution?

2. How is $Z_T$ used for guidance ,which is mentioned in section 3.4.2?

---

> ### Author Response · Authors · 2024-11-23
> **Rebuttal by Authors**
>
> > Weakness 1: The layout design of Figure 2 is unappealing and does not effectively highlight the core innovations of this research.
>
> Response: Thanks for the valuable suggestion. The revised Figure 2 is included in the re-submitted manuscript. In the new figure, 3 steps (encoding history, predicting goals, predicting next steps) are explicitly illustrated. A more detailed description is also added to the caption of the figure.
>
> > Weakness 2: what does $q$ in the Formula 7 and Formula 8 represent for and what is the meaning of $i$, $j$ and $k$?
>
> Response: In formula 8, $q$ is the center coordinate for each cell of the grid. In these formulas, $i$ is the index of pedestrian, which is from 1 to $N$; $j$ and $k$ are the row and column indexes of the cell in the whole grid. Then coordinate $q$ is computed by the row and column index and the size of each cell. Then in the next step, $q$ is sent to coordinate feature encoding module (Formula 7).
>
> > Questions 1: In Formula 11, what is the prior distribution of $\epsilon$, and does $p(\epsilon)$ represent the standardized normal distribution?
>
> Response: Yes. Formula 11 is a standard CVAE loss function, where the distribution of random variable $\epsilon\sim p(\epsilon)$ is assumed to be a standardized normal distribution.
>
> > Questions 2: How is $Z_T$ used for guidance ,which is mentioned in section 3.4.2?
>
> Response: As shown in Figure 2, $Z_T$ is sent to step encoder in every step prediction procedure to provide the knowledge of where the goal is. (Figure 2 will be updated to make it clearer.)

---

> > ### Comment · Reviewer_RVzK · 2024-11-26
> >
> > I read the responses, which partly solve my concerns. However, the original paper presents some confusing points and weakness as other reviewers commented. I will keep the score.

---

> > > ### Author Response · Authors · 2024-11-27
> > > **Response to Reviewer RVzK**
> > >
> > > Thank you for taking the time to respond to the authors' rebuttal. We are pleased to hear that our rebuttal has addressed some of your concerns. We would greatly appreciate it if you could share more details about any remaining concerns from your original comments, as well as any new concerns after reading other reviewers’ comments and authors’ responses. With the discussion deadline extended, we would like to know your detailed concerns and further discuss them.

---

### Official Review · Reviewer_RsFQ · 2024-11-01

**Soundness:** 2
**Presentation:** 3
**Contribution:** 2
**Rating:** 6
**Confidence:** 4

**Summary:**

This paper addresses the scene usage limitations of prior approaches in pedestrian trajectory prediction by introducing the trajectory-scene-cell (TSC) feature and TSC-Net. The authors convert raw trajectory and scene inputs into three types of TSC features: trajectory cells, goal scene cells, and step scene cells, which are utilized for trajectory prediction. The paper reframes the trajectory prediction task as a classification problem. Furthermore, the authors claim that their method achieves state-of-the-art performance and is demonstrated better
on predicting goals for trajectories with irregular speed.

**Strengths:**

1.	The trajectory-scene-cell feature in pedestrian trajectory prediction is somewhat innovate.
2.	The paper is well-written and easy to follow.

**Weaknesses:**

1.	My main concern is whether the TSC feature genuinely works better than the traditional CNN and heatmap approaches. Because in the performance comparison, TSC-Net significantly underperforms compared to Y-Net, NSP-SFM, V2-Net, and E-V2-Net. And there is no ablation study provided to demonstrate the performance of replacing the TSC feature with traditional CNN and heatmap approaches.
2.	The authors claim that Y-net and NSP-SFM performs better because their results come from clustering 1000 samples. However, I only find they also use best-of-20 (K=20) protocol in their paper. Moreover, if they indeed use extra samples, the result of the TSC-Net under the same setting could be given for comparison.
3.	The authors claim in abstract that TSC-Net is demonstrated better on predicting goals for trajectories with irregular speed. However, Figure 3 only provides a few examples for demonstration, which is insufficient to illustrate the overall situation. A more detailed statistical table should be included.
4.	The ETH-UCY and SDD datasets are relatively small and consist of simple trajectories. The performance of TSC-Net on more complex datasets, such as NBA[1], would provide a clearer understanding of its capabilities.
[1] Conghao Wong, Beihao Xia, Ziqian Zou, Yulong Wang, and Xinge You. Socialcircle: Learning the angle-based social interaction representation for pedestrian trajectory prediction. In Proceedings of the IEEE/CVF Conference on Computer Vision and Pattern Recognition, pp. 19005–19015, 2024.

**Questions:**

1.	Explain the underperformance of all four approaches in ETH-UCY and demonstrate TSC feature genuinely works better than the traditional CNN and heatmap approaches.
2.	A more detailed statistical table should be included for irregular speed as author claims.
3.	Performance on more complex datasets, such as NBA[1].
Other issues can refer to the weaknesses.

[1] Conghao Wong, Beihao Xia, Ziqian Zou, Yulong Wang, and Xinge You. Socialcircle: Learning the angle-based social interaction representation for pedestrian trajectory prediction. In Proceedings of the IEEE/CVF Conference on Computer Vision and Pattern Recognition, pp. 19005–19015, 2024.

---

> ### Author Response · Authors · 2024-11-23
> **Rebuttal by Authors**
>
> > Weakness 1 and Questions 1: Whether the TSC feature genuinely works better than the traditional CNN and heatmap approaches. And there is no ablation study provided to demonstrate the performance of replacing the TSC feature with traditional CNN and heatmap approaches. Explain the underperformance of all four approaches in ETH-UCY and demonstrate TSC feature genuinely works better than the traditional CNN and heatmap approaches.
>
> Response: First, our TSC feature is not a plug-and-play module, nor is it a feature that can be straightforwardly replaced by a CNN feature or a heatmap. As explained in line 315 in our manuscript, TSC is a combination of coordinate encoding and scene encoding, which achieves the alignment of two different feature types. The prediction framework discussed in Section 3.4 is derived from our TSC feature, making it infeasible to replace the TSC feature with a CNN feature or a heatmap.
>
> The CNN feature usually encodes the global scene as a vector, where the correspondence between coordinates and the scene around specific coordinates is lost. Heatmaps retain all spatial correspondences but cannot model pedestrian-pedestrian relations due to their computational complexity. Our TSC feature is capable of modeling both pedestrian-pedestrian relations and maintaining spatial correspondence to some extent.
>
> Our method focuses on modeling the relations between the scene and pedestrians as well as among pedestrians. The SDD dataset, with its complex scene semantic maps, highlights the strengths of our method. As described in Table 2, our method outperforms all other methods. Without scene maps, predicting trajectories in this grid manner may not be as natural as coordinate auto-regression, yet our method still achieves comparable results.
>
>
> > Weakness 2: Different sample numbers in Y-Net and NSP-SFM.
>
> Response: Y-net (NSP-SFM follows the same goal prediction framework) uses the best-of-20 metric. However, the 20 samples are obtained from clustering 10,000 samples from the heatmap (described as the Test-Time Sampling Trick in their supplementary material), which naturally covers larger possible areas of the scene.
>
> Moreover, a comparison under the same setting is not practicable. Sampling 10,000 times in a heatmap is straightforward in Y-Net, whereas in our framework (and most of other methods with generation framework), sampling 10,000 times would requires going through the CVAE inference framework 10,000 times. This process is too time-consuming and cannot be completed within a limited time.
>
>
> > Weakness 3 and Questions 2: A more detailed statistical table should be included for irregular speed as author claims.
>
> Response: Thanks for the valuable suggestion. Here, we define a velocity difference between history and future $V_{diff}$. Velocity is the absolute value of the Euclidean distance between two neighboring frames $v_t=||p_{t+1}-p_{t}||_2$. The history mean velocity is defined as the average velocity of all historical frames. Similarly, the future mean velocity is defined as the average velocity of all future frames. The velocity difference is then the absolute difference between the history and future velocities, where
>
> $V_{diff}=|\frac{\sum_{t=1}^{\tau}v_t}{\tau}-\frac{\sum_{t=\tau+1}^{T}v_t}{T-\tau}|$.
>
> We compute the ADE/FDE with different velocity differences. For example in SDD dataset with long-term prediction setting, for the 10% of samples with the largest velocity difference, our performance is 85.41/101.86, while Y-Net is 100.10/162.50. A more detailed comparison with the velocity difference distribution is included in the revised supplemental materials due to the page limitation of main paper.
>
>
> > Weakness 4 and Questions 3: Performance on more complex datasets, such as NBA[1].
>
> Response: We evaluate our method on NBA dataset with 10-frame prediction setting, where the resutls are shown as follow.
> | Methods | ADE/FDE |
> |----------|----------|
> | E-V$^2$-Net (2023)   | 1.26 / 1.64   |
> | MemoNet (2022)   | 1.25 / 1.47   |
> | V$^2$-Net-SC (2024)   | 1.22 / 1.51   |
> | E-V$^2$-Net-SC (2024)   | 1.18 / 1.46   |
> | Ours  |  1.24 / 1.50   |
>
> Our method does not perform [1] but still achieves comparable performance when comparing to other methods in Table 2 in [1]. One possible reason is that trajectories in the NBA dataset are highly influenced by player-player interactions, as players need to react to each other’s movements, while the scene has little influence. Our method is designed to address feature alignment between the scene and trajectory, and its capability could be limited when scenes are excluded. The resutls are added to the supplimenal pdf file.

---

### Official Review · Reviewer_iGai · 2024-11-02

**Soundness:** 3
**Presentation:** 3
**Contribution:** 3
**Rating:** 8
**Confidence:** 4

**Summary:**

The paper proposes a novel framework, TSC-Net, with an innovative Trajectory-Scene-Cell (TSC) feature representation designed for pedestrian trajectory prediction.  In particular, it effectively captures both individual pedestrian behavior and the contextual scene information. By decoupling trajectory data in the temporal domain and scene information in the spatial domain, the model effectively captures both human-human interactions and human-scene dynamics, leading to improved prediction accuracy compared to existing methods. The experiments conducted on various datasets seem to validate the efficacy of the proposed method.

**Strengths:**

+ The paper is well written and easy to follow, with clear explanations of methodologies, results, and implications.

+ It is a well-designed idea to amalgamate trajectory and scene information into a unified TSC feature space. Such a joint feature allows for better modeling of both human-human and human-scene interactions.

+ The authors reformulate the trajectory prediction task as a classification problem by estimating the likelihood of each cell containing the goal or the next steps, which distinguish the proposed methods from the existing regression-based trajectory prediction.

+ Extensive experimentation across multiple datasets, i.e., SDD and ETH-UCY, demonstrating the robustness and generalizability of TSC-Net. Such a clear improvement over state-of-the-art methods strongly demonstrate the model's efficacy.

**Weaknesses:**

- The proposed model has shown tendencies to bias its trajectory predictions towards the centers of scene cells. This could lead to inaccuracies, particularly when pedestrians navigate along edges or less central paths.

- The discussion of limitations is relatively brief. If the authors can provide some vivid examples, then it should be easier for readers to better understand.

- It is still not clear if the proposed method is able to sufficiently address scenarios with higher pedestrian densities or complex environments especially in urban settings with numerous dynamic agents.

**Questions:**

See the "Weakness" section.

---

> ### Author Response · Authors · 2024-11-23
> **Rebuttal by Authors**
>
> > Weakness 1 & 2: The proposed model has shown tendencies to bias its trajectory predictions towards the centers of scene cells. The discussion of limitations is relatively brief. If the authors can provide some vivid examples, then it should be easier for readers to better understand.
>
> Response: Thanks for the suggestion about the limitation discussion. As we discussed in line 525 in the paper, all the grids are centered on each pedestrian. It means that if the pedestrian has no movement, the goal prediction will be located at the center point of the central cell in the grid (with zero offset). With a non-negligible number of pedestrians without movement in the training set, this could bias the offset prediction.
>
> Moreover, we compute the trajectory length distribution on SDD dataset with long-term prediction setting, where about 18% of the trajectory whose length is shorter than 1/10 of the cell edge size. To solve this problem, it requires further analysis and may requires an additional step to do the movement/non-movement classification in the future.
>
> > Weakness 3: It is still not clear if the proposed method is able to sufficiently address scenarios with higher pedestrian densities or complex environments especially in urban settings with numerous dynamic agents.
>
> Response: A high-density scenario could be a task with different challenging points, as the relation modeling among different pedestrians has an O(N^2) complexity. With limited computational resources, the range of relation modeling needs to be carefully determined instead of including every pedestrian in consideration. Furthermore, interaction modeling may requires a specific movement simulation module to explicitly model the social interaction rules. We will carefully consider the possible solutions towards high-density scenario in our future works.

---

> > ### Comment · Reviewer_iGai · 2024-11-28
> >
> > Thank you for the response. I would like to keep my original rating but meanwhile I agree there are still room for the authors to further polish the current paper with the other reviewers' feedback.

---

### Author Response · Authors · 2024-11-23
**Rebuttal by Authors**

We appreciate the valuable comments and suggestions provided by all the reviewers. We are grateful for the time and effort all the reviewers dedicated to our manuscript. Please refer to our individual responses for detailed information on each review. We look forward to further comments from reviewers. The manuscript and pdf file in supplemental material are also revised accordingly and re-submitted.

---

### Meta-Review · Area_Chair_QHdZ · 2024-12-17

**Metareview:**

The paper proposes a novel framework, TSC-Net, with an innovative Trajectory-Scene-Cell (TSC) feature representation designed for pedestrian trajectory prediction. The paper is well written and easy to follow, with clear explanations of methodologies, results, and implications. Extensive experimentation across multiple datasets, i.e., SDD and ETH-UCY, demonstrating the robustness and generalizability of TSC-Net. Such a clear improvement over state-of-the-art methods strongly demonstrate the model's efficacy. This paper lacks a discussion of the limitations of the proposed methods, especially for complex dynamic scenarios. The reviewers also pointed out the lack of experiments to verify the advantages of the method over traditional methods, the unclear description of the motivation and the lack of comparison with the latest methods. The author's detailed response can address most of the reviewer's concerns. So the final vote is acceptance.

**Additional Comments On Reviewer Discussion:**

See Metareview.

---

### Decision · Program_Chairs · 2025-01-22

Accept (Poster)